# Formulation Challenges and Strategies to Develop Pediatric Dosage Forms

**DOI:** 10.3390/children9040488

**Published:** 2022-04-01

**Authors:** Wedad A. Malkawi, Enas AlRafayah, Mohammad AlHazabreh, Salam AbuLaila, Abeer M. Al-Ghananeem

**Affiliations:** 1College of Art and Science, University of Louisville, Louisville, KY 40208, USA; wedad.malkawi@louisville.edu; 2College of Pharmacy, Jordan University of Science and Technology, Irbid 22110, Jordan; eialrafayah17@ph.just.edu.jo (E.A.); mkalhazabreh17@ph.just.edu.jo (M.A.); ssabulaila17@ph.just.edu.jo (S.A.); 3College of Pharmacy and Health sciences, Sullivan University, Louisville, KY 40205, USA

**Keywords:** pediatrics, dosage form, formulation development, oral, buccal, regulatory, excipients

## Abstract

The development of pediatric-specific dose forms is particularly difficult due to a variety of factors relating to pediatric population differences from adult populations. The buccal dosage form is considered a good alternative to oral dosage form if the latter cannot be used in pediatric patients. Both oral and buccal dosage formulations uphold great application qualities for pediatric patients. This review sheds light on both oral and buccal, as they are the most convenient dosage forms for pediatrics. The use of adult drugs to treat children is a legislation concern, as it may result in incorrect dose, safety, and efficacy. The Best Pharmaceuticals for Children Act (BPCA) and the Pediatric Research Equity Act (PREA) are two key pieces of legislation that encourage and regulate pediatric medication research. Both contribute to a well-balanced approach to emphasizing critical safety and efficacy warnings for the of medications within pediatric populations. These contributions are what enable companies to continue making significant investments in pediatric drug developments. Despite the importance of investigating medicines for children, there is still a demand for pediatric-specific formulations and dosage forms. Many formulations and dosage forms can be designed, among which the buccal drug delivery seems a good modality for pediatric-friendly dosage forms. The main issues associated with these pediatric dosage forms development, particularly clinical and physiological factors, are discussed in this review. In addition, formulation developments and regulatory expectations are highlighted. In turn, suggestions are made to potentially improve future pediatric formulation development.

## 1. Introduction

Pediatric patients have traditionally been treated with off-label adult pharmaceuticals or extemporaneous compounding, which is not optimal for pediatric patients due to a lack of product safety and efficacy. In addition, off-label drug usage becomes much more complicated when it comes to neonates and children under the age of two. This is also applicable to children with rare or chronic disorders or acutely ill children. New drugs are frequently approved for adults without even a minuscule quantity of expertise in pediatric patients. Due to the absence of information, practitioners are commonly obliged to prescribe medications off-label, using poorly defined dosage methods. Half the medicine in the United States alone is not labeled for use among children [1]. Over one-third of pediatric hospitalizations (36.7%) were associated with off-label use of analgesics that have not been labeled for use in children [2].

The Best Pharmaceuticals for Children Act (BPCA) and the Pediatric Research Equity Act (PREA) are aimed to encourage more pediatric medication studies. The major objectives of these regulations include increasing in the number of medications clinically tested for pediatric use as well as ensuring these medications are available to the market with appropriate formulations and doses [3]. In addition to promoting the development of pediatric drugs, there are other peculiarities for pediatric patients that must be considered. As a result, many therapeutic medicines continue to lack pediatric-friendly dosage forms. Although the legislative decisions provided by the BCPA and the PREA benefit the pediatric research of new drugs and their labeling, these mandates can be outweighed by medications previously on the market. Medications made before the application of the legislative regulations provided by the BCPA and PREA are prone to not benefiting from these efforts.

It can be challenging to construct pediatric formulations, especially those suitable for young infants. There is a lack of data on the acceptability of different dosage forms, administration volume, dosage form size and taste, and the safety of formulation excipients in relation to age and development status. Moreover, factors such as dosing, variances in disease processes, study design, and placebo response are all linked to failures in pediatric trials [4]. It is critical to understand the reasons for failed studies and consider new strategies to design better pediatric trials in the future. Wharton et al. reported that pediatric labeling was not established for 78 medications out of 189 products under pediatric exclusivity (1998–2012), which is a failure rate of 42% [5].

The oral route tends to be most favorably used in pediatric formulations. Oral pediatric formulations are oftentimes provided in powder forms for reconstitution. These forms require purified water for reconstitution and sometimes require unique storage conditions in a refrigerated environment; yet these requirements cannot always be met. Liquid and powder forms may be unpalatable and difficult to swallow, which makes them problematic for pediatric patients. Children’s swallowing abilities may vary from adults’; however, it is usually hard for children to swallow solid dosage forms. Children younger than five years old are generally unable to safely swallow solid capsules and tablets larger than 10 mm [6]. It must also be taken into consideration that children may reject medication regardless of tablet size if the flavor is unpleasant. Clarithromycin, for example, is characteristically bitter [7] and has poor palatability when compared to other antibiotics. As a result, medications with similar unpleasantness regarding flavoring may cause conflict with dose scheduling and adherence. In place of the presence of vomiting or nausea via the oral route, despite being less common, the rectal route may be used as an alternative [8]. There is a multitude of rectal dosage forms present within markets, including suppositories, creams, enemas, and ointments; yet, administering them can most often be uncomfortable for children. In case of emergencies or oral route difficulty, the parenteral route may be adopted due to its speed. Despite the advantage of speed, the parenteral route comes with many limitations, such as the need for a trained professional to administer, the invasiveness of the process, the risk of blood-borne infections, and the presence of injury or pain induced by injections [9]. To achieve the rapid onset of action and high bioavailability goals in treatment, clinicians tend to choose the parenteral route. This approach has several drawbacks, including patient acceptability, sterility limits, and the availability of medical experts to administer. Regarding the listed challenges, the buccal route is far more appealing. In addition, the buccal route can be much more suitable in cases such as seizures, where the oral, rectal, or parenteral routes are all limited.

In addition, pediatric patients with cancer suffer from severe breakthrough episodes of pain and are in need of prompt and efficient pain treatment. The oral route, in this situation, is a poor choice, as it may take a prolonged period before eliminating discomfort. However, it could be of benefit for maintenance treatment of pain to have a sustained-release effect of medication over a prolonged period. Buccal medication delivery is an appealing administration route for pediatric pain management since it has a quick onset of action and has no hepatic first-pass metabolism. Although oral and buccal dosage formulations uphold great application qualities for pediatric patients in comparison to other commonly used dosage forms, there are still many challenges related to those dosage forms, which can be summarized into three major areas related to physiological considerations, regulatory expectations, and formulation development consideration (Figure 1).

## 2. Physiological Considerations

Pediatric patients are defined as patients aged from birth to less than 16 or 18 years old (Table 1) [10]. The classification is dependent on the individual’s age. However, categorizing pediatric patients into distinct age groups for drug administration is essentially arbitrary, as it does not consider pharmacokinetic parameters, such as renal or hepatic function, volume of distribution, lipophilicity, relative blood volumes, etc. This is an issue, as certain formulations are age-specific or must be given according to body weight. As a result, the acceptability of different dose forms for different age groups must be carefully considered.

The following are some of the clinical and physiological issues to consider while developing buccal pediatric dose forms:

### 2.1. Heterogeneity among Pediatric Patients

Variability among children (0–17 years) is a significant challenge in developing age-appropriate pediatric-friendly formulations, as this group differs in taste preferences, medicine-related side effects, and patient medical case—all of which can influence doses and dosage forms required [10]. Pediatric patients have different responses to the active ingredients and excipients due to changes in their body physiology.

### 2.2. Age-Related Compliance

Age-related compliance is an important challenge to consider in formulation design. Compliance is complicated, and sometimes it is caregiver dependent, especially when considering the younger ages and varying disease states. Pediatric patients have different cognitive levels during their growth, which will affect compliance for formulation. There is also a dependency on caregivers to play a role in the administration of the designed formula [11]. Age-appropriate formulations must be utilized to enhance the taste preferences of the children who are accommodated [12].

### 2.3. Saliva-Flow Rate

For oral and oral transmucosal dosage forms, the presence of saliva is imperative for drug absorption. Saliva provides a relatively aqueous environment to facilitate drug release, dissolution, and absorption. Over the life span of a human, both the saliva-flow rate and composition tend to change [13]. The flow rate tends to increase up until around 5 to 6 years, and then, the saliva flow rate declines, whereas the mean electrolyte content rises [14].

The rate of saliva secretion in the buccal mucosa was found to be significantly lower in children (0.22–0.82 mL/min) than in adults (0.33–1.42 mL/min) [15,16,17].

There is a direct correlation between the degree of hydration via saliva and the objectionable drug swallowing. Premature medication swallowing, on the other hand, may occur in patients with a high saliva-flow rate (the “saliva washout effect”). This action may result in non-uniform medication dispersion in saliva as well as decreased drug absorption by mucosal tissues, resulting in wide-ranging systemic bioavailability [18].

### 2.4. Gastrointestinal Tract (GIT) pH Values and Drug Absorption

The pH of the GIT varies depending on the location of the tissue and the patient’s age. It has a significant impact on the solubility and diffusion of drug forms, preferring the unionized form. Other factors that impact the pH of the oral mucosa include disease, drug use, nutrition, and saliva-flow rate. In general, as the rate of saliva-flow increases, so does the pH [19,20]. The intra-gastric pH of newborns is higher than 4, and intestinal cytochrome p450 activity increases with age [21]. The mean pH value of the oral mucosa is (6.78 ± 0.04) and (6.64 ± 0.44) in healthy adults and pediatric patients, respectively [19,20].

There is also significant variability in the pharmacology of drug and dosage forms between adult and pediatric populations. This is due to differences in the absorption, allocation, metabolic activity, and excretion (ADME) profile in pediatric populations, particularly within a child’s first few years [22].

## 3. Formulation Development Considerations

The development of pediatric-friendly formulations and dosage forms can be challenging due to specific nuance for this patient population. One of the main challenges in the development of pediatric dosage forms is the selection of the most appropriate formulation in relation to patient age. Furthermore, one should consider the dosing regimen (dose accuracy, flexibility, frequency, etc.), route of administration, dosage form, ingredients compatibility and stability, suitability and regulatory aspect of excipient, and patient compliance. The latter is of considerable effect due to the known noncompliance in the pediatric population due to swallowability and palatability for oral and buccal dosage forms [23].

Tablets and capsules are typically not suited for children under the age of four, and available tablets for older children may not be of suitable strength. Although splitting or breaking tablets is widespread, it does not lead to accurate or consistent dosing [24].

Oral liquids, such as solutions and suspensions, are the most widely utilized dose forms for pediatric patients. Lozenges, gummies, chewing gums, and lollipops are other frequent buccal dosage forms. Figure 2 illustrates the different oral and transmucosal buccal dosage forms. It can be noted that extemporaneous made liquid formulations are an alternative when commercial liquid formulations are unavailable. Pediatric formulation compounding, particularly for newborns, can be complex for dispensation. The absence of suitable doses of various medications is a challenge when treating ill neonates. In U.S. hospitals, the preparation of oral medicines for children varies greatly, as there is little standardization for formulas or preparation information on stability.

When commercial pediatrics dosage forms are unavailable, bulk powders and occasionally tablets can be utilized as a drug source in formulations. However, knowledge of formulation components is crucial to ensure making a safe and effective pediatric formulation. For example, one can use lorazepam bulk powder as a drug source to make oral suspensions for pediatric patients, but one cannot use commercial lorazepam injections as a drug source for pediatrics due to the high propylene glycol contents. Excipients such as propylene glycol can certainly be harmful to neonates especially in high dosages [25,26].

Furthermore, it is important to know the additives and excipients in tablets before using them as a drug source to make oral solutions or suspensions. It can thicken the solution or suspension. Therefore, the concentration might have to be diluted to reach the appropriate rheology.

### 3.1. Dosage Form Selection

After determining the dose, one of the most essential considerations in the establishment of oral pediatric pharmaceuticals is whether they are appropriate for the target patient age group without causing issues in palatability or swallowing. The key component impacting patient compliance is most presumably taste. Children may dislike the bitter or metallic taste of unmasked formulations and even dislike the flavor of masked formulations. They might also refuse doses with large volume. Flavor, sweetener, pH, color, and tongue feel all contribute to a formulation’s taste. Furthermore, dose, stability, preservation, packaging, and the requisite administration devices or techniques are among variables to consider while designing pediatric formulations. When considering buccal dosage forms, it is important to know the physicochemical properties of the drug molecule to evaluate its suitability for buccal absorption. To ensure high bioavailability of a drug, it is ideally to have a solubility greater than 1 mg/mL, molecular size lower than 500 Da, lipophilicity with a log P greater than 10 and less than 1000, and be in the unionized form in buccal pH [27,28]. For this reason, the pKa value of a drug becomes an important factor, as it determines the degree of ionization of a drug at different pH values, which also affect drug solubility, absorption, and bioavailability [9,29].

Oral liquid formulations, such as solutions and suspensions, are the most ideal oral formulation for children because they are easier to swallow. They are prepared by solubilizing or suspending the drug source in appropriate aqueous carriers [30]. However, liquid dosage forms intended for buccal delivery have the disadvantage of being difficult to hold in the oral cavity when it comes to pediatric patients, and swallowing may occur before transmucosal absorption. This can release a large amount of medication in an unregulated manner throughout the oral cavity. Therefore, a need for research in pediatric-friendly dosage forms is needed to assess the safety of pharmaceutical technologies, including the use of mucoadhesive polymers, that could overcome the short contact time issue.

Drug solubility may limit the dose and volume of liquid medicines utilized. Co-solvents or surfactant excipients may be required in the formulation to address this issue. Sweeteners and flavors may be used to hide an unpleasant taste. If masking taste is not possible, a more complex formulation strategy, such as drug particle encapsulation, can be used [31]. However, all this needs to be studied on pediatric patients, as their bodies respond to excipients in a different manner compared to adults.

Effervescent buccal discs are planar and thinner than buccal tablets or conventional effervescent tablets, which result in a rapid drug release and hopefully a fast onset of action. Carbon dioxide is one of the essential ingredients that is released from buccal effervescent discs upon contact with saliva, acting as a permeation enhancer. Although this is an attractive alternative for pediatric patients due to increased bioavailability, further investigations are required to assess the safety of formulation ingredients in this dosage form [32].

Lozenges and lollipops are solid dosage forms prepared to dissolve or disintegrate slowly in the mouth. They comprise one or more drugs usually in a flavored and sweetened formulation. Patients should put the dosage form between the cheek and gum or suck on it to start the drug release and absorption. This form has the advantages of being easy to administer to pediatric patients, having formulas that are easily changed to be patient specific, and offering the ability to keep the drug in contact with the oral cavity for an extended period. Due to the frequency and intensity of suction, dissolution and disintegration of the drug contained in lozenges and lollipops can result in an increase in uncontrollable swallowing. This may lead to unexpected changes in drug pharmacokinetics [33].

Films, wafers, and strips are also used as buccal delivery systems that release the drug directly towards the buccal epithelium and have gained relevance in the pharmaceutical industry, as they are considered patient friendly and easy to administer. However, their limited dimensions may cause a huge challenge in the maximum amount of drug that can be incorporated [34]. Table 2 lists examples of oral transmucosal formulations and their excipients. There is a tremendous need for more pediatric friendly dosage forms to so many other medications.

### 3.2. Excipient Selection

Since what is safe for adults is not always the case for pediatric patients, there are specific safety considerations for any excipient included in pediatric formulations. The safety of utilized excipients play a huge role in excipient choice for pediatric formulations. Collaboration between pharmaceutical experts is vital in the development of a good design as well as determining the maximum number of excipients, concentrations, and best route for administration [33]. Literatures and databases, such as STEP (Safety and Toxicity of Excipients for Pediatrics), can be of great benefit in early stages of developing pediatric formulations. When evaluating the safety profile for excipients, one should take into consideration the daily exposure and expected duration of treatment for the specific age group of pediatric patients.

#### 3.2.1. Preservatives

Preservatives are needed in aqueous multi-dose preparations but not as much in solid dosage preparations. The minimum preservative concentration level with satisfactory anti-microbial function must be used in children’s formula [35]. While necessary to use in some pharmaceutical products, these agents were associated with significant adverse effects in pediatric patients [36]. “Gasping baby syndrome” is one of the serious adverse reactions related to the use of preservatives. Table 3 lists different type of preservatives and potential side effects related to their use. In general, it is recommended to avoid preservative containing products especially benzyl alcohol in infants whenever possible [37].

#### 3.2.2. Taste-Masking Agents

There are different taste receptors distributed on the papillae of the tongue and throughout the oral cavity. Molecules, after being dissolved in saliva, will interact with taste receptors, and children can recognize these tastes very well, as they have a well-developed sensory system [38]. The unpleasant taste for active pharmaceutical ingredients can be masked by adding flavor or sweetening agents. The penchant for good taste in pediatric patients is highly affected by their experiences and their cultures, which must be considered when developing pediatric dosage form to ensure acceptability and compliance [39].

There are many innovative pharmaceutical technologies that have emerged to mask taste of medications (Figure 3). However, these technologies and their ingredients and solvents would need to be evaluated for safety in pediatric patients.

#### 3.2.3. Sweeteners

Sucrose is one of the commonly used sweetening agents in pharmaceutical formulation, as it rapidly hydrolyzed in the intestine. However, considerable amounts of fructose and glucose could be released by mucosa bounded enzymes and absorbed through the lumen [40]. In pediatric patients, it is better to replace sucrose with a sucrose-free formula (i.e., aspartame, glycerol, or other alternatives), as sucrose can cause dental cavities and dissolve tooth enamel in children [41]. Most importantly, sucrose should be avoided in pediatric patients having diabetes as well as those who have an intolerance to fructose. Hill et al. reported that among 160 oral liquid and chewable medications of pediatric formulations investigated, the sucrose concentration in some could go up to 80% and that only four antibiotics among the ones investigated were sucrose free [42].

Aspartame is known as an artificial sweetener (150–200 times sweeter than sucrose). It is a di-peptide of aspartic acid and phenylalanine ester, making it harmful for pediatrics with phenylketonuria [43]. Consuming large amounts of aspartame make it neurotoxic, as it may induce seizures. Therefore, it is important that the proper amount of aspartame used be labeled [44]. Aspartame artificial sweetener could be replaced by stevia, date sugar, maple sugar, maple syrup molasses, and agave nectar in pediatric formulations.

In young infants, sorbitol accumulation can also lead to diabetic complications, such as retinopathy and cataracts. Therefore, the amount of sorbitol is limited to 0.3 gm/kg in pediatric formulations.

Glycerol is used in oral formula as a solvent and mild sweetener, as it has hygroscopic and osmotic properties. Small amounts of glycerol posture no adverse effect; yet, large amounts will cause electrolyte disturbances resulting in diarrhea [43]. Table 3 lists different type of sweeteners and potential side effects related to their use.

#### 3.2.4. Coloring Agents

Pediatrics prefer colored preparations [45]. Despite providing positive influences in preference or ability in identifying products, coloring agents should be avoided unless necessary [43]. Most regulatory agencies around the world restrict the use of coloring agents, such as tartrazine, since the azo dyes have been linked to hypersensitivity reactions and ADHD in children [46]. Azo dyes should be avoided in pediatric formulation, as there is high concern for coloring agents with natural sources. Despite going through an extraction process, these naturally sourced coloring agents tend to still leave traces of proteins responsible for allergic reactions [47]. Such dyes can be substituted with vegetable dyes, such as annatto, malt beta-carotene, and turmeric, or not used at all in pediatric formulations. Table 3 lists different type of coloring agents and potential side effects related to their use.

#### 3.2.5. Mucoadhesive Agents

Mucoadhesive polymers used in oral and buccal formulations tend to increase contact time and drug-delivery rate at the site of absorption [32,48]. Poly(acrylic) acid derivatives work by forming a hydrogen bond with mucin and are the most prevalent mucoadhesive used in formulation [49]. Polyvinylpyrrolidone is another mucoadhesive agent with the advantage of stability in large pH changes. Safety studies are needed to better assess the use of these innovative pharmaceutical technologies and agents in pediatric patients.

#### 3.2.6. Penetration Enhancers

Also known as permeation or absorption enhancers, penetration enhancers work by increasing the rate of absorption through the biological membrane by increasing the drug portioning in the oral epithelium as well as increasing drug retention time on the buccal mucosa surface [32]. Surfactants and bile salts are great examples of penetration enhancers, yet they still need to be evaluated for safety in pediatric patients.

**Table 3 children-09-00488-t003:** Pharmaceutical excipients and potential adverse effects related to their use [29,30,36,38,50,51,52,53].

Excipients	Excipient and Potential Adverse Effects
Preservatives	Benzyl alcohol and sodium benzoic acid: Gasping syndrome. Methylparaben and polyparaben: Kernicterus. At high doses, it may cause hyperbilirubinemia and oestrogenicity. It should be avoided in pediatric patients with jaundice. Polysorbate 80: E-Ferol syndrome. Ethanol: Central nervous system (CNS) depression.
Sweeteners	Sucrose: Dental cavities and dissolving of tooth enamel. Sorbitol: Gastro intestinal (GI) disorders, such as abdominal pain, bloating, vomiting, and diarrhea, when used in high concentrations. In young infants, sorbitol accumulation can also lead to diabetic complications, such as retinopathy and cataracts. Aspartame: Harmful for pediatrics with phenylketonuria. Neurotoxic, as it may induce seizures. Glycerol: In high concentration, will cause electrolyte disturbances, resulting in diarrhea.
Coloring agents	Potential Attention-deficit disorder (ADHD) effect Most regulatory agencies around the world restrict the use of coloring agents, such as tartrazine, since the azo dyes have been linked to hypersensitivity reactions and ADHD in children. Such dyes can be substituted with vegetable dyes, such as annatto, malt beta-carotene, and turmeric, or not used at all in pediatric formulations.
Enteric coating polymers	Phthalates play a critical role as a coating agent (plasticizer) in modified-release formulations. Exposure of phthalates to the fetus has been linked to developmental abnormalities, such as cleft palate and skeletal malformations and fetal death.
Diluents	Microcrystalline cellulose: Potential intestinal absorption; should not be used in children < 2 years. Lactose: Hypersensitivity reactions in children and young infants. In infants with lactose intolerance (lactase deficiency), it could cause severe abdominal pain, flatulence, distention or bloating, and diarrhea in addition to systemic symptoms, such as muscle and joint pain, and eczema can occur. It can be substituted with calcium hydrogen phosphate dehydrate, starch, powdered cellulose, and erythritol.
Cosolvents	Alcoholic solvents, such as sorbitol, propylene glycol, polyethylene glycol (PEG), and others; could cause CNS depression, hypoglycemia, lactic acidosis, seizure, hypoglycemia, and hemolysis.

## 4. Regulatory Considerations

Thoughtful drug development and inclusion of pediatric patients in clinical trials is crucial to develop pediatric-friendly dosage forms and for public health. If the drug is not intended for pediatric use, then health care providers would have two options: to treat with medications based on adult studies with limited pediatric experience (off-label use) or not to treat pediatric patients with potentially beneficial medications because the medications were not approved for use in pediatric patients.

Currently, one may see early integration in the development of new drugs due to the enforcement by the regulatory agencies such as the Food and Drug Administration (FDA). In the USA, the pediatric drug development laws are governed by Pediatric Research Equity Act (PREA), which requires companies to assess safety and effectiveness of new drugs/biologics in pediatric patients and by Best Pharmaceuticals for Children Act (BPCA), which provides a financial incentive to companies to voluntarily conduct pediatric studies.

Initial investigations of a pediatric formulation are usually conducted in adults to demonstrate acceptable bioavailability. Once the formulation’s safety and efficacy in adults is established, clinical trials in children are conducted. Cautious and adequate planning is needed when conduction pediatric clinical trials due to the wide variability among age groups of pediatric patients and their vulnerability. If no adequate clinical investigation is commenced, several medical issues and adverse effects in pediatric patients can become a major problem.

Furthermore, when designing the clinical study protocol, a suitable therapeutic outcome measuring tool must be developed for pediatric patients of different ages. The research procedures and setting should take into consideration ethical and regulatory standards in addition to the child’s cognitive, physical, and emotional development [54].

Clinical trials in pediatric patients could be problematic sometimes due to the small number of children with the same medical condition. Additionally, it could be difficult to discern a parent’s decision to involve their children in research. To complicate the situation, sometimes pharma companies strive to create novel oral dosage forms, such as ones with a unique release profile. However, these dosage forms might not be suitable to produce certain concentrations for pediatrics or may contain excipients that cannot be used in pediatric patients. This complicates the regulatory requirements and makes it unsuitable to be tested in pediatric patients. In certain cases, the novel dosage form might be in sizes that cannot be swallowed by children. Therefore, pharma companies might request from FDA a partial PREA waiver, as the unique tablet can be produced in a size that cannot be swallowed young age pediatric patients. A partial PREA waiver will be appropriate only after reasonable attempts to produce a pediatric formulation necessary for that age group have failed, which adds to cost and timeline exasperations.

Therefore, it is crucial to think early on to design pediatric-friendly dosage forms to be in compliance with PREA regulatory requirements.

## 5. Conclusions

Unlike adult pharmaceutical formulations, pediatric dosage developments tend to be a challenge. This is due to the significant difference in pediatric pharmacotherapy. These elemental distinctions include the preference of taste, ability to self-administer medications, drug-related toxicities, and others.

Additionally, pediatric patients tend to show variable responses to active ingredients due to changes in body physiology. This includes membrane permeability, enzymatic activities, pharmacokinetics, and pharmacodynamics. Despite the numerous difficulties faced during pediatric formulation, the oral route is the most convenient approach to administer drugs. This is due to the ability of the oral route to adhere to all safety and satisfactory measures among young patients. In turn, these abilities shine light on buccal administrations and formulations, which are advantageous for being a simple and convenient route for drug administration among enticing dose forms.

Additionally, formulation scientists must be adaptable in the choice of dosage forms and excipients. This is to develop formulations that meet the needs of patients while also accommodating the properties of the drug. Although pharmaceutical excipients are an important component to drug formulation, it is crucial to remember that these compounds can induce adverse effects. That said, most patients tolerate inert ingredients without issue. The rising availability of labeling information and the tendency toward more cautious usage of these substances will aid health care practitioners in selecting appropriate pediatric-friendly products.

Although there are legislative decisions provided by the BCPA and PREA to benefit pediatric research, formulation scientists and pharma companies must conduct additional research to create pediatric-friendly dosage forms. That said, drugs developed prior to the adoption of BCPA and PREA legislations are unlikely to benefit from these efforts. This can be addressed by implementing an amendment to provide better incentives to the pharma industry for drugs already in the market, thus increasing the likelihood of more pediatric-friendly formulations becoming available.

In conclusion, further steps are necessary to develop pediatric-friendly dosage forms and improve the health, clinical care, and overall well-being of pediatric patients.

## Figures and Tables

**Figure 1 children-09-00488-f001:**
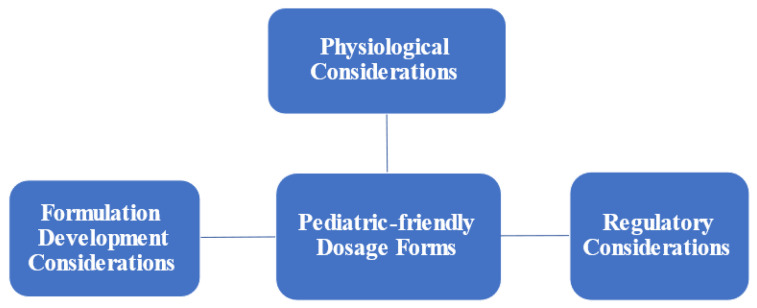
Challenges associated with developing pediatric-friendly dosage forms.

**Figure 2 children-09-00488-f002:**
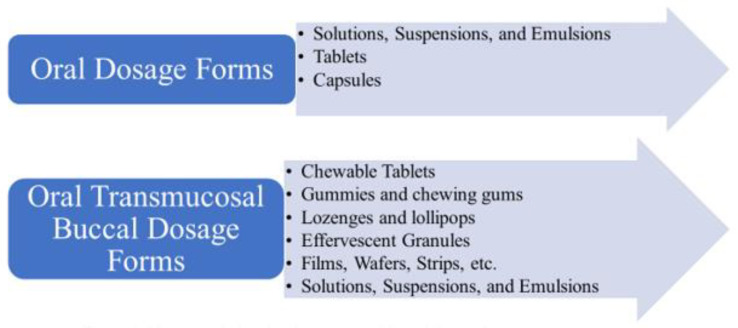
Pharmaceutical oral and transmucosal buccal dosage form.

**Figure 3 children-09-00488-f003:**
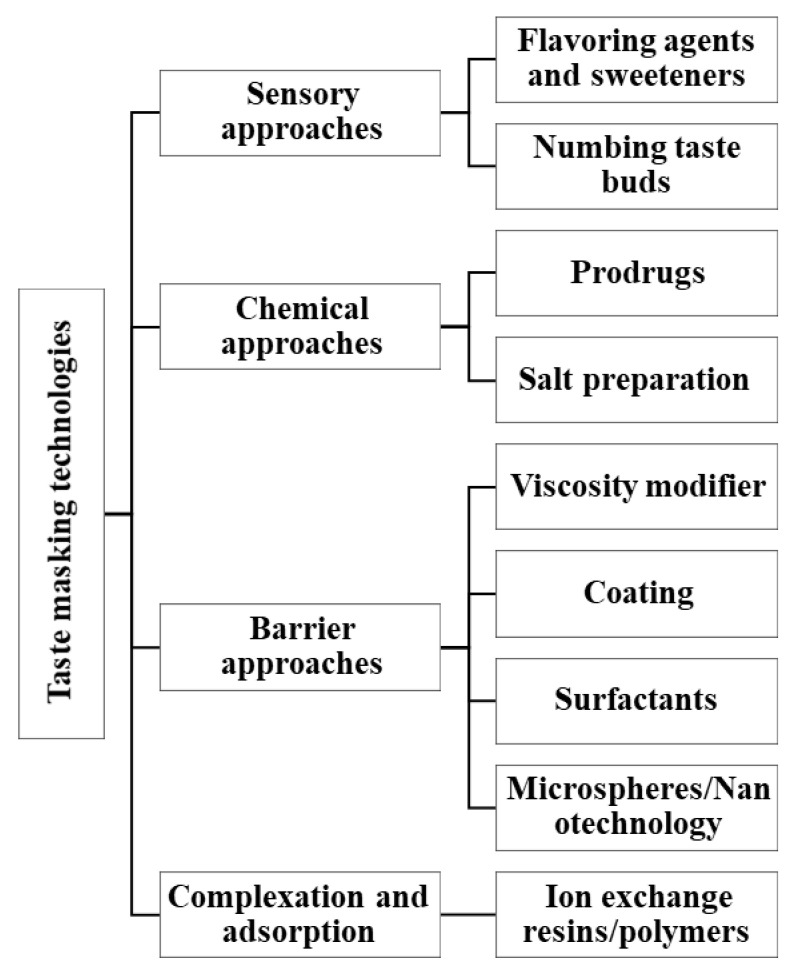
Pharmaceutical technology techniques for taste masking that would need to be assessed for safety in pediatric formulations.

**Table 1 children-09-00488-t001:** Classification of pediatric age categories [10].

Class	Age Group
Neonate	Birth to 27 days
Infant and Toddlers	28 days to 23 months
Children	2 to 11 years
Adolescent	12 to 16–18 years

**Table 2 children-09-00488-t002:** Examples of oral transmucosal buccal formulations and their excipients.

Active Ingredient	Dosage Form	Indication	Examples of Additives in the Dosage Form	Approved Age
Diphenhydramine HCL Benadryl^®^	Chewable tablet	Anti-allergic	crospovidone, D&C red no. 30 aluminum lake, D&C red no.7 calcium lake, dextrose excipient, ethylcellulose, FD&C blue no.1 aluminum lake, flavors, gum arabic, hydroxypropylcellulose, magnesium stearate, microcrystalline cellulose, sucralose, sugar spheres, tartaric acid	Above 6 years
Buprenorphine Temges^®^	Sublingual tablet	Pain management	lactose, mannitol, maize starch, povidone K30, citric acid anhydrous, magnesium stearate, sodium citrate, purified water and ethanol (96%)	Above 6 years
Midazolam BUCCOLAM^®^	Buccal solution	Antiseizure	sodium chloride, water for injections, hydrochloric acid and sodium hydroxide	Above 3 months

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
