# Peer review of "Formulation Challenges and Strategies to Develop Pediatric Dosage Forms"

_children, 2022, doi:10.3390/children9040488_

Round 1

Reviewer 1 Report

Developing dosage forms suitable for the pediatric population is challenging and there is still a great need to develop more age appropriate formulations. Thus, this is an interesting topic. However, there are several published reviews in this area and it is a bit difficult to see the added value of this review to the already existing literature. What gap in knowledge is filled with the review? There are relatively few references for a review and references are missing in several places where statements should be supported by previous publications/studies. Furthermore, there are few newly published references.

The focus on buccal administration is not clear. The paper also includes oral administration and dosage forms but why not only focus on buccal administration and dosage forms? I believe the paper would benefit from this. When different oral dosage forms is discussed, there is no mention of for example mini tablets which can be considered a suitable dosage form to children.

Line 36-37: Could you please clarify the sentence ”New drugs are frequently approved without even a minuscule quantity of expertise in pediatric patients.” Where in the process is the expertise missing?

Line 46: drug delivery methods - consider using dosage forms

Line 67: from what age does children normally accept to swallow tablets? Children is a heterogeneous group consisting of large difference in age. A 1 year old would have problem with swallowing tablets but perhaps not a 12 year old.

Line 93-94: is this in relation to oral and buccal dosage forms? Please clarify.

Line 105-106. the focus on buccal administration is not clearly motivated. Why only include aspects of buccal administration if the paper also includes oral administration?

Line 117: could you please clarify what you mean by age-appropriate communications and how this affect taste preferences

Line 134: pH also affects solubility of the drug

Line 140-143: references are missing

Line 188-189: the suitability and regulatory aspects of excipients also need to be considered

Line 190-191: references are missing

Line 194-195: references are missing

Line 232: pH also affects solubility of the drug

Line 288: reference is missing

Line 302-304: references are missing

Figure 3 needs to be enlarged

Line 331: reference is missing 

Table 3 needs references

The conclusion is too long and should focus on the main findings.

Author Response

Reviewer 1:

Developing dosage forms suitable for the pediatric population is challenging and there is still a great need to develop more age appropriate formulations. Thus, this is an interesting topic. However, there are several published reviews in this area and it is a bit difficult to see the added value of this review to the already existing literature. What gap in knowledge is filled with the review? There are relatively few references for a review and references are missing in several places where statements should be supported by previous publications/studies. Furthermore, there are few newly published references.

Thank you. We updated and added more references as suggested.

The focus on buccal administration is not clear. The paper also includes oral administration and dosage forms but why not only focus on buccal administration and dosage forms? I believe the paper would benefit from this. When different oral dosage forms is discussed, there is no mention of for example mini tablets which can be considered a suitable dosage form to children.

The authors wanted to shed light on both oral and buccal as they are the most convenient dosage forms for pediatrics. This review will be a great reference for someone wanted to learn more about challenges and strategies to develop such dosage forms.

Line 36-37: Could you please clarify the sentence ”New drugs are frequently approved without even a minuscule quantity of expertise in pediatric patients.” Where in the process is the expertise missing?

We modified the sentence to clarify the meaning:

” New drugs are frequently approved for adults without even a minuscule quantity of expertise in its effect on pediatric patients.”

Line 46: drug delivery methods - consider using dosage forms. Done!

Line 67: from what age does children normally accept to swallow tablets? Children is a heterogeneous group consisting of large difference in age. A 1 year old would have problem with swallowing tablets but perhaps not a 12 year old.

We added the following statement to complete the idea and clarify the age and swallowing matter.

Children younger than five years old are generally unable to safely swallow solid capsules and tablets larger than 10 mm [Ref].

Line 93-94: is this in relation to oral and buccal dosage forms? Please clarify.

Yes, we added the following to clarify the sentence:

“Although oral and buccal dosage formulations uphold great application qualities for pediatric patients in comparison to other commonly used dosage forms, but there are still many challenges, related to those dosage forms, which can….”

Line 105-106. the focus on buccal administration is not clearly motivated. Why only include aspects of buccal administration if the paper also includes oral administration?

Usually, the buccal dosage form is considered a good alternative to oral dosage form if the later can’t be used in pediatric patients.  Both oral and buccal dosage formulations uphold great application qualities for pediatric patients. The authors wanted to shed light, in this review, on both oral and buccal as they are the most convenient dosage forms for pediatrics. This review will be a great reference for someone wanted to learn more about challenges to develop such dosage forms.

Line 117: could you please clarify what you mean by age-appropriate communications and how this affect taste preferences

Sorry, this was a typo. It should be “formulations” not “communications”. It is now corrected in the manuscript.  

Line 134: pH also affects solubility of the drug

Correct! It was added as suggested see below:

The pH of the GIT varies depending on the location of the tissue and the patient's age. It has a significant impact on the solubility and diffusion of drug forms, preferring the unionized form.

Line 140-143: references are missing

Van den Anker JN, Schwab M, Kearns G. Pediatric Clinical Pharmacology. Handbook of Experimental Pharmacology Eds: HW Seyberth et al., (Springer-Verlag Berlin Heidelberg). 2011:51-75.

Line 188-189: the suitability and regulatory aspects of excipients also need to be considered

Done! It was added as suggested see below:

Furthermore, one should consider the dosing regimen (dose accuracy, flexibility, frequency, etc.), route of administration, dosage form, ingredients compatibility, stability, suitability and regulatory aspect of excipient, and patient compliance.

Line 190-191: references are missing

Boateng J. Drug delivery innovations to address global health challenges for pediatric and geriatric populations (through improvements in patient compliance). Journal of Pharmaceutical Sciences. 2017;106(11):3188-98.

Line 194-195: references are missing:

van Riet-Nales DA, Doeve ME, Nicia AE, Teerenstra S, Notenboom K, Hekster YA, et al. The accuracy, precision and sustainability of different techniques for tablet subdivision: breaking by hand and the use of tablet splitters or a kitchen knife. International Journal of Pharmaceutics. 2014;466(1-2):44-51.

Line 232: pH also affects solubility of the drug

Done! It was added as suggested see below:

“… of a drug at different pH values, which also affect drug solubility, absorption, and bioavailability.”

Line 288: reference is missing  

Cuzzolin L. Neonates exposed to excipients: concern about safety. Journal of Pediatric and Neonatal Individualized Medicine (JPNIM). 2018;7(1):e070112-e

Line 302-304: references are missing 

Technologies were graphed and listed based on authours’ experience and knowledge.

Figure 3 needs to be enlarged. Done!

Line 331: reference is missing 

Walsh LM, Toma RB, Tuveson RV, Sondhi L. Color preference and food choice among children. J Psychol. 1990 Nov;124(6):645-53.

Table 3 needs references. We listed references as requested.  

The conclusion is too long and should focus on the main findings.

The authors shortened the conclusions as requested.

Reviewer 2 Report

  • This paper touches on an important topic in pediatrics, and one that is difficult to narrow in the scope of a review. I saw only one instance of self-citation, and it does not appear to negatively impact the paper in any way. While this is an important topic, the overall sense of this paper is that it offers little in the way of solutions or actionable items for a practitioner or scientist. In that sense, who is the target audience? The paper vacillates somewhat between targeting practitioners and scientists.
  • The manuscript calls attention to several interesting points surrounding growth and development, and it would be great to expand upon these. For example: 2.3. Saliva Flow Rate and 2.4. Gastroinstestinal Tract pH Values and Drug Absorption are both highly data-driven and, if expanded, may serve as models for additional sections.
  • The mention of legislature may be better incorporated into the introductory section as a lead-in to the topic. 
  • Overall, the paper reads as a bit disorganized. Topics are mentioned, and then mentioned again at later points. There are also many instances of "editorializing" without referenced support. Rather than pointing out specific sentences, I will offer that any claims other than summary statements are likely to require a reference in a review paper. 
  • There are several sections that use definitive language that lumps "pediatric" patients together when the section would be better served by use of stratified age categories (i.e. 4.2.1 Preservatives).
  • Title: The paper appears to consider less strategies and more challenges; may consider rewording.
  • Abstract: Contains instances of editorializing (ex: "major source of worry") and appears to overstate role of legislature in context of paper. 
  • Line 35 - suggest changing "unusual" to "rare"
  • Has the statistic pulled from reference one been updated in the last 10 years? This is an older reference for this type of data. 
  • Lines 53 - 55 appear not to connect to each other. 
  • Line 56 - 57: Is this what was done in reference 3? If so, consider rewording this sentence. 
  • Line 63: implies that both liquid and powder formulations need reconstitution.
  • Line 64: not all reconstitutables require refrigeration
  • Second paragraph, page 2: What is the heart of this paragraph? It appears to mention many things without being able to be distilled to a main point. 
  • Lines 86 - 87: is this referring to breakthrough episodes of pain?
  • Line 88: are there instances where the oral route may be preferred?
  • Figure 1 repeats information within the text; suggest removing.
  • Line 100: States "essentially arbitrary," however, this does not take into account renal or hepatic function, Vd, lipophilicity, relative blood volumes, etc.
  • Reference for table 1?
  • I am unfamiliar with infant classification extending to two years of age.
  • Inconsistency noted between table classification of "adolescent" ages and other portions of text
  • Section 2.2. Compliance is complicated in that for the younger ages and varying disease states, it is caregiver dependent. Compliance does not seem like the correct concept here, despite the fact that caregiver dependency is addressed.
  • Section 3 seems repetitive to what was previously stated. 
  • Line 178: suggest defining what type of waiver is being referred to here
  • Figure 2 appears to be referencing dosage forms most commonly used in adults. For example, prochlorperazine is not approved in children younger than 2 years of age; zolpidem is not approved for use in children. Based on these types of examples, why are these agents (for example) being shown in this table? This table may be misleading in that a provider may misinterpret these agents as safe to use in children. Additionally, this needs referencing as excipients are proprietary-specific.
  • Section 4: first-person language is used here; consider re-wording; also consider lower-case for all generic agents
  • Line 211: suggest specifying dosages of concern, rather than making a blanket statement
  • Line 218: previous section was discussing the concept of "excipients;" how does this transition to "dose?" 
  • Page 6, first paragraph: volume may also be considered a deterrent.
  • Section 4.2.2.: interesting mention of culture-related flavor profiles! 
  • Figure 3: too small to assess. I was unable to read this table. 
  • Line 310: I am unclear as to what is recommended when making a switch from sucrose to sucrose-free formulations, based on the commentary that follows. Is there a statement that could summarize all of the suggestions? 
  • Line 312: suggest rewording "pediatric diabetics" into patient-first language
  • Line 331: What is this based off of? 
  • Table 3: Consider inserting references
  • Conclusions section: Do lines 360 - 361 contradict the statements made in the abstract that buccal formulations are good options? 

Author Response

Reviewer 2:

  • This paper touches on an important topic in pediatrics, and one that is difficult to narrow in the scope of a review. I saw only one instance of self-citation, and it does not appear to negatively impact the paper in any way. While this is an important topic, the overall sense of this paper is that it offers little in the way of solutions or actionable items for a practitioner or scientist. In that sense, who is the target audience? The paper vacillates somewhat between targeting practitioners and scientists.

Thank you! The authors intention of this review is to be among the references to be used by a wide range of audiences including practitioners and scientists.

  • The manuscript calls attention to several interesting points surrounding growth and development, and it would be great to expand upon these. For example: 2.3. Saliva Flow Rate and 2.4. Gastroinstestinal Tract pH Values and Drug Absorption are both highly data-driven and, if expanded, may serve as models for additional sections.

We agree on the importance of these points, and we believe they were adequately covered within the scope of the review article.

  • The mention of legislature may be better incorporated into the introductory section as a lead-in to the topic. 

The legislation is already incorporated in the introductory section at page 2.

  • Overall, the paper reads as a bit disorganized. Topics are mentioned, and then mentioned again at later points. There are also many instances of "editorializing" without referenced support. Rather than pointing out specific sentences, I will offer that any claims other than summary statements are likely to require a reference in a review paper. 

The authors added reference support as suggested. Furthermore, the article systemically lists challenges and strategies in pediatric formulations. We hope this is obvious in in the updated version.

  • There are several sections that use definitive language that lumps "pediatric" patients together when the section would be better served by use of stratified age categories (i.e. 4.2.1 Preservatives).

The authors tried to specify the age categories when applicable. Section 4.2.1 already specifies infants without referring to pediatrics.

  • Title: The paper appears to consider less strategies and more challenges; may consider rewording.

The article shed light at both strategies and challenges. If the reviewer feels there are less strategies and more challenges, then we are changing the order of challenges and strategies in the title to read as follow:

Formulation Challenges and Strategies to Develop Pediatric Dosage Forms

  • Abstract: Contains instances of editorializing (ex: "major source of worry") and appears to overstate role of legislature in context of paper. 

The statement was rewritten to clarify the point. It is now:

The use of adult drugs to treat children is a legislation concern a major source of worry, as it may result in incorrect dose,

  • Line 35 - suggest changing "unusual" to "rare". Done!

  • Has the statistic pulled from reference one been updated in the last 10 years? This is an older reference for this type of data. 

We could not locate updated statistics; but we added another statement from a newer reference to support the idea. See below.

Half the medicine in the United States alone is not labeled for use among children [1]. Over a third of pediatric hospitalizations (36.7%) were associated with off-label use of analgesics that have not been labeled for use in children [Carmack, 2020].

Carmack M, Berde C, Monuteaux MC, Manzi S, Bourgeois FT. Off-label use of prescription analgesics among hospitalized children in the United States. Pharmacoepidemiol Drug Saf. 2020 Apr;29(4):474-481. doi: 10.1002/pds.4978. Epub 2020 Feb 26. PMID: 32102118

  • Lines 53 - 55 appear not to connect to each other. 

The statements were revised. It read now as:

There is a lack of data on the acceptability of different dosage forms, administration volume, dosage form size, taste, and the safety of formulation excipients in relation to age and development status due to failures in conducting pediatric trials [3]. Factors such as dosing, variances in disease processes, study design, and placebo response are all linked to failures in pediatric trials [3].

  • Line 56 - 57: Is this what was done in reference 3? If so, consider rewording this sentence. 

The paragraph is a suggestion from the authors. It reads now as below:

It is critical to understand the reasons for failed studies and consider new strategies to design better pediatric trials in the future.

  • Line 63: implies that both liquid and powder formulations need reconstitution.

To minimize confusion, we deleted liquid and kept powder forms.

Oral pediatric formulations are oftentimes provided in liquid or powder forms for reconstitution.

  • Line 64: not all reconstitutables require refrigeration

We revised the statement, and it reads now as follows:

These forms require purified water for reconstitution and sometimes require as well as unique storage conditions in a refrigerated environment

  • Second paragraph, page 2: What is the heart of this paragraph? It appears to mention many things without being able to be distilled to a main point. 

The paragraph is discussing challenges to construct pediatric formulations and trials in small pediatric population and the need to design better pediatric trials.

We hope the revised submission is clearer.

  • Lines 86 - 87: is this referring to breakthrough episodes of pain?

Yes! The statement was revised as follow:

…patients with cancer suffer from severe pains, breakthrough episodes of pain, and the…

  • Line 88: are there instances where the oral route may be preferred?

Yes! We added the following to show when the oral route could be preferred:

The oral route, in this situation, is a poor choice as it may take a prolonged period before eliminating discomfort. But it could be of benefit for maintenance treatment of pain to have a sustained release effect of medication over a prolonged period.

  • Figure 1 repeats information within the text; suggest removing.

The authors prefer to keep it as it could help visual readers and instate the information provided.

  • Line 100: States "essentially arbitrary," however, this does not take into account renal or hepatic function, Vd, lipophilicity, relative blood volumes, etc.

Agree! The statement was revised as follow:

The classification is dependent on the individual's age. However, categorizing pediatric patients into distinct age groups for drug administration is essentially arbitrary as it does not consider pharmacokinetic parameters such as renal or hepatic function, volume of distribution, lipophilicity, relative blood volumes, etc. This is an issue as certain formulations are age-specific or must be given according to body weight. As a result, the acceptability of different dose forms for different age groups must be carefully considered.

  • Reference for table 1?

Standing JF, Tuleu C. Paediatric formulations – getting to the heart of the problem. Int J Pharm. 2005 Aug 26;300(1–2):56–66. doi: http://dx.doi. org/10.1016/j.ijpharm.2005.05.006 PMID: 15979830

  • I am unfamiliar with infant classification extending to two years of age.

We revised the literature and chose a classification that is more popular. We updated Table 1 content and added a reference accordingly.

  • Inconsistency noted between table classification of "adolescent" ages and other portions of text

After updated the table it should be consistent now.

  • Section 2.2. Compliance is complicated in that for the younger ages and varying disease states, it is caregiver dependent. Compliance does not seem like the correct concept here, despite the fact that caregiver dependency is addressed.

Agree! We added the following to emphasize the point raised by the reviewer:

Age-related compliance is an important challenge to consider in formulation design. Compliance is complicated and sometimes it is caregiver dependent, especially when considering the younger ages and varying disease states.

  • Section 3 seems repetitive to what was previously stated. 

Section 3 emphasizes the regulatory consideration in a concise way easy for the reader to follow.

  • Line 178: suggest defining what type of waiver is being referred to here

It was clarified in the revised submission to read as follow:

from FDA a partial PREA waiver as the unique tablet can be produced in a size that cannot be swallowed young age pediatric patients. A partial PREA waiver will

  • Figure Table 2 appears to be referencing dosage forms most commonly used in adults. For example, prochlorperazine is not approved in children younger than 2 years of age; zolpidem is not approved for use in children. Based on these types of examples, why are these agents (for example) being shown in this table? This table may be misleading in that a provider may misinterpret these agents as safe to use in children. Additionally, this needs referencing as excipients are proprietary-specific.

We revised table 2 and added the approved age to appropriate examples.

  • Section 4: first-person language is used here; consider re-wording; also consider lower-case for all generic agents

Lower case was used for generic agents.

  • Line 211: suggest specifying dosages of concern, rather than making a blanket statement

The statement was specified as follow:

Excipients such as propylene glycol It can certainly be harmful to neonates …

  • Line 218: previous section was discussing the concept of "excipients;" how does this transition to "dose?" 

Previous section introduces “formulation development consideration”. Then listing items related to this major point starting with 4.1 Dosage Form Selection.

  • Page 6, first paragraph: volume may also be considered a deterrent.

Thank you! We added a statement to reflect effect of volume.

The key component impacting patient compliance is most presumably taste. Children may dislike the bitter or metallic taste of unmasked formulations and even dislike the flavor of masked formulations. They might also refuse doses with large volume.

  • Section 4.2.2.: interesting mention of culture-related flavor profiles! Correct! It is interesting.

  • Figure 3: too small to assess. I was unable to read this table. It was enlarged in the revised submission.

  • Line 310: I am unclear as to what is recommended when making a switch from sucrose to sucrose-free formulations, based on the commentary that follows. Is there a statement that could summarize all of the suggestions? 

To clarify the idea choices of sucrose-free options were added as follow:

sucrose with sucrose-free formula (i.e., aspartame, glycerol, or other alternatives) as sucrose can cause ….

  • Line 312: suggest rewording "pediatric diabetics" into patient-first language

Kindly see updated statement:

Most importantly, sucrose should be avoided in pediatric patients having diabetes as …

  • Line 331: What is this based off of? Reference was added.

  • Table 3: Consider inserting references. Done!

  • Conclusions section: Do lines 360 - 361 contradict the statements made in the abstract that buccal formulations are good options? 

Although it is a good modality to deliver medications, but pediatric dosage form development is a challenge as described in the article. We revised the conclusion to better clarify the idea.  

Round 2

Reviewer 1 Report

I thank the authors for their revised manuscript. They have addressed several of my comments. However, I still have some concerns/comments and have listed them below.

Line 36-37: In the answers to my comments it is mentioned that the sentence ”New drugs are frequently approved without even a minuscule quantity of expertise in pediatric patients” has been revised but this is not done in the manuscript.

The authors respond that ”Usually, the buccal dosage form is considered a good alternative to oral dosage form if the later can’t be used in pediatric patients. Both oral and buccal dosage formulations uphold great application qualities for pediatric patients. The authors wanted to shed light, in this review, on both oral and buccal as they are the most convenient dosage forms for pediatrics.”

I believe that something similar this should be stated in the paper as well so that the starting point is clear to the reader.

Line 203-204: ”Although splitting or breaking tablets is widespread, it does not lead to accurate or consistent dosing” 

The authors should support this with more than one reference.

Line  220-221: reference regarding the harmfulness of propylene glycol is missing

Regarding excipients and children there is a recently published review on that topic: Rouaz, K.; Chiclana-Rodríguez, B.; Nardi-Ricart, A.; Suñé-Pou, M.; Mercadé-Frutos, D.; Suñé-Negre, J.M.; Pérez-Lozano, P.; García-Montoya, E. Excipients in the Paediatric Population: A Review. Pharmaceutics 2021, 13, 387

Line 312: has is missing the sentence ie ”has been”

Although the authors state that this is based on their experience and knowledge I still feel that some references are appropriate regarding taste masking technologies.

Table 3: ”relate” should be changed to ”related”

Line  369-371: The sentence is not clear and should be rephrased …”their significant difference in pharmacotherapy”, what is ”their” a reference to?

The conclusions needs to be rewritten. When just delting text the context is partly missing.

For example, line 382, ”these ability” is reference to the previous sentence that now has been deleted

Line 392, ”these substances” is reference to the previous sentence that now has been deleted

Author Response

03/21/2022

Reviewer 1:

I thank the authors for their revised manuscript. They have addressed several of my comments. However, I still have some concerns/comments and have listed them below.

Line 36-37: In the answers to my comments it is mentioned that the sentence ”New drugs are frequently approved without even a minuscule quantity of expertise in pediatric patients” has been revised but this is not done in the manuscript.

Thank you! It is now corrected in the revised manuscript.

” New drugs are frequently approved for adults without even a minuscule quantity of expertise in its effect on pediatric patients.”

The authors respond that ”Usually, the buccal dosage form is considered a good alternative to oral dosage form if the later can’t be used in pediatric patients. Both oral and buccal dosage formulations uphold great application qualities for pediatric patients. The authors wanted to shed light, in this review, on both oral and buccal as they are the most convenient dosage forms for pediatrics.”

I believe that something similar this should be stated in the paper as well so that the starting point is clear to the reader.

Thank you for the beneficial suggestion. We added the following to the abstract to ensure the starting point is clear to the reader.

“The buccal dosage form is considered a good alternative to oral dosage form if the later can’t be used in pediatric patients.  Both oral and buccal dosage formulations uphold great application qualities for pediatric patients. This review shed light on both oral and buccal as they are the most convenient dosage forms for pediatrics."

Line 203-204: ”Although splitting or breaking tablets is widespread, it does not lead to accurate or consistent dosing” 

The authors should support this with more than one reference.

Sure! We added the two references below:

Hill SW, Varker AS, Karlage K, Myrdal PB. Analysis of drug content and weight uniformity for half-tablets of 6 commonly split medications. J Manag Care Pharm. 2009 Apr;15(3):253-61. doi: 10.18553/jmcp.2009.15.3.253. PMID: 19326956.

Freeman MK, White W, Iranikhah M. Tablet splitting: a review of weight and content uniformity. Consult Pharm. 2012 May;27(5):341-52. doi: 10.4140/TCP.n.2012.341. PMID: 22591978.

Line  220-221: reference regarding the harmfulness of propylene glycol is missing

MacDonald MG, Getson PR, Glasgow AM, et al. Propylene glycol: increased incidence of seizures in low birth weight infants. Pediatrics. 1987;79(4):622–625.

 Allegaert K, Vanhaesebrouck S, Kulo A, et al. Prospective assessment of short-term propylene glycol tolerance in neonates. Arch Dis Child. 2010;95(12):1054–1058.

Regarding excipients and children there is a recently published review on that topic: Rouaz, K.; Chiclana-Rodríguez, B.; Nardi-Ricart, A.; Suñé-Pou, M.; Mercadé-Frutos, D.; Suñé-Negre, J.M.; Pérez-Lozano, P.; García-Montoya, E. Excipients in the Paediatric Population: A Review. Pharmaceutics 2021, 13, 387

Line 312: has is missing the sentence ie ”has been”

Thank you! Corrected.  

… pharmaceutical technologies that have been emerged …

Although the authors state that this is based on their experience and knowledge I still feel that some references are appropriate regarding taste masking technologies.

Sure! We added the following reference to figure 3.

Sohi H, Sultana Y, Khar RK. Taste masking technologies in oral pharmaceuticals: recent developments and approaches. Drug Dev Ind Pharm. 2004 May;30(5):429-48. doi: 10.1081/ddc-120037477. PMID: 15244079

Table 3: ”relate” should be changed to ”related”

Thank you! Done!

Line  369-371: The sentence is not clear and should be rephrased …”their significant difference in pharmacotherapy”, what is ”their” a reference to?

In reference to pediatrics.

The conclusions needs to be rewritten. When just delting text the context is partly missing.

The conclusion was revised accordingly.

For example, line 382, ”these ability” is reference to the previous sentence that now has been deleted

Line 392, ”these substances” is reference to the previous sentence that now has been deleted

Thank you! The conclusion was revised accordingly.

Reviewer 2 Report

It is clear that many changes and revisions were made to this paper - kudos to the authors for a swift review and turnaround. Rather than provide a line-by-line review, I would like to make some general statements on the paper as a whole. 

The paper touches on a topic that has significance to the pediatric community. Dosage formulation is an important and significant topic to practitioners and scientists. 

Overall, this paper provides many challenges, and few actionable items; however, more importantly, it seems that it is not clearly focused on the thesis of what it seeks to do. The paper appears to want to focus on legislature and buccal formulation issues and challenges. If this is the case, my suggestions would be:

-revising the title and abstract to reflect this

-narrowing the scope of the introduction to condense into a "needs statement" for these items. As is, the introduction contains many items that may be better served in the body of the paper. 

-restructure the paper so that legislation comes either first or at the end, rather than in-between the subsections, although I'm not entirely sure why the legislature is completely necessary for a paper that focuses on buccal formulation issues and challenges; these seem like they could be two separate topics altogether.

-revisit items included as tables to be sure that serve have a necessary role in the paper and are additive to the discussion, and do not serve as unnecessary distractors from the topic at hand, which is, in my opinion, buccal formulation challenges and strategies

-conduct another revision to remove unnecessary adjectives and editorializing, review for changes in tense, and remove first-person language.

Author Response

03/21/2022

Reviewer 2:

It is clear that many changes and revisions were made to this paper - kudos to the authors for a swift review and turnaround. Rather than provide a line-by-line review, I would like to make some general statements on the paper as a whole. 

Thank you! True, the authors worked with enthusiasm to provide responses to your comments which were valuable.

The paper touches on a topic that has significance to the pediatric community. Dosage formulation is an important and significant topic to practitioners and scientists. 

Thank you! We concur!

Overall, this paper provides many challenges, and few actionable items; however, more importantly, it seems that it is not clearly focused on the thesis of what it seeks to do. The paper appears to want to focus on legislature and buccal formulation issues and challenges. If this is the case, my suggestions would be:

The authors wanted to shed light, in this review, on both oral and buccal as they are the most convenient dosage forms for pediatrics. The legislation was mentioned to give a complete picture on pediatric formulation development from a review article prospective.

-revising the title and abstract to reflect this

We already revised the title in the first-round of revisions and made challenges before strategies.

We now added a paragraph in the abstract to shed light on the focus of the review article regarding buccal and oral formulations.

-narrowing the scope of the introduction to condense into a "needs statement" for these items. As is, the introduction contains many items that may be better served in the body of the paper. 

The authors feel that the introduction is part of the body of the review article, and it provides needed info to complete the idea to the reader.

-restructure the paper so that legislation comes either first or at the end, rather than in-between the subsections, although I'm not entirely sure why the legislature is completely necessary for a paper that focuses on buccal formulation issues and challenges; these seem like they could be two separate topics altogether.

As requested, the authors moved the legislation section to the end. The authors believe that this is what makes this review unique. The legislation was mentioned to give a complete picture on pediatric formulation development from a review article prospective. It is a driving force for pushing toward producing more pediatric dosage forms.

-revisit items included as tables to be sure that serve have a necessary role in the paper and are additive to the discussion, and do not serve as unnecessary distractors from the topic at hand, which is, in my opinion, buccal formulation challenges and strategies

Thank you! As requested, we revisited the manuscript, and we believe the items available are of benefit to the cause.  

-conduct another revision to remove unnecessary adjectives and editorializing, review for changes in tense, and remove first-person language.

Thank you! As requested, we conducted another round of revisions and editorializing as seen in track change mode in the revised manuscript.
